# Mobilome of Apicomplexa Parasites

**DOI:** 10.3390/genes13050887

**Published:** 2022-05-16

**Authors:** Matias Rodriguez, Wojciech Makalowski

**Affiliations:** Institute of Bioinformatics, Faculty of Medicine, University of Münster, 48149 Münster, Germany; matidae@gmail.com

**Keywords:** transposons, Apicomplexa, bioinformatics, mobile elements

## Abstract

Transposable elements (TEs) are mobile genetic elements found in the majority of eukaryotic genomes. Genomic studies of protozoan parasites from the phylum Apicomplexa have only reported a handful of TEs in some species and a complete absence in others. Here, we studied sixty-four Apicomplexa genomes available in public databases, using a ‘de novo’ approach to build candidate TE models and multiple strategies from known TE sequence databases, pattern recognition of TEs, and protein domain databases, to identify possible TEs. We offer an insight into the distribution and the type of TEs that are present in these genomes, aiming to shed some light on the process of gains and losses of TEs in this phylum. We found that TEs comprise a very small portion in these genomes compared to other organisms, and in many cases, there are no apparent traces of TEs. We were able to build and classify 151 models from the TE consensus sequences obtained with RepeatModeler, 96 LTR TEs with LTRpred, and 44 LINE TEs with MGEScan. We found LTR Gypsy-like TEs in Eimeria, Gregarines, Haemoproteus, and Plasmodium genera. Additionally, we described LINE-like TEs in some species from the genera Babesia and Theileria. Finally, we confirmed the absence of TEs in the genus Cryptosporidium. Interestingly, Apicomplexa seem to be devoid of Class II transposons.

## 1. Introduction

Most eukaryotic genomes are populated by a myriad of interspersed repetitive sequences that originated from mobile genetic elements known as transposable elements (TEs). Due to their mobile nature and their ability to multiply within the genome, TEs have successfully populated the vast majority of eukaryotic genomes to such a degree that in many organisms they constitute the bulk of the genome. The proliferation of TEs has a great impact on gene and genome evolution as it can impair genes, modify gene expression, disrupt regulatory sequences, facilitate chromosome rearrangements, and is one of the main factors behind genome expansion, and thus contributes to the generation of evolutionary novelties [1].

In protozoan parasites, an important characteristic of their genomes is the extreme plasticity that can be observed as large polymorphisms in homologous chromosomes and extensive genomic rearrangements that can be partially attributed to the activity of TEs [2]. Despite the importance of TEs in shaping genomes and as drivers of genome evolution, the identification of these elements can be a very difficult task due to their great diversity and high degree of sequence divergence.

The phylum Apicomplexa comprises a large number of protozoan organisms which are obligate intracellular parasites of a wide range of vertebrate and invertebrate hosts [3]. They are responsible for multiple pathologies of medical and veterinary importance, including malaria, toxoplasmosis, and cryptosporidiosis in humans, eimeriosis in poultry, and theileriosis in cattle. Interestingly the discovery of transposons in apicomplexan parasites proved to be a difficult task, even if some species of Plasmodium were very well studied.

To date, only a handful of TEs have been identified in Apicomplexa species. For instance, in RepBase [4], one of the most commonly used databases of repetitive sequences from eukaryotic organisms, there are only three well characterized TEs in Apicomplexa, each of them from different Eimeria species, seven unknown repetitive elements from *Plasmodium falciparum*, two from *Toxoplasma gondii*, and one from *Theileria parva*. Additionally, protein coding sequences with signatures of transposable elements were identified in Plasmodium species using a consensus sequence of retrotranscriptase [5].

In avian malaria genomes, the presence of transposable elements in the Plasmodium lineage was reported, the majority as fragments of LTR-retrotransposons in *Plasmodium gallinaceum* and *Plasmodium relictum* encoding domains belonging to the Gypsy family TE [6]. Moreover, in *Ascogregarina taiwanensis,* retrotransposable elements belonging to the Gypsy family were reported and ORFs encoding for gag-pol polyprotein [7]. When the chromosome 1 of *Eimeria tenella* was studied, the possible presence of transposons was reported [8], and later, as more Eimeria species were sequenced, the presence of highly divergent and fragmented LTR transposons was confirmed [9].

Apicomplexans have intricate origins [10] and their genomes are complex and extremely divergent [11]. Their genomes’ analyses are challenging due to the extreme GC composition and the organisms’ multiple adaptations to parasitism [12]. The genome size can vary immensely from approximately 8 to 10 Mb for species from the genus Cryptosporidium to 124 Mb for *Sarcocysits neurona*, and also the GC content has high variations, from 17% in some species of Plasmodium to 57% in Besnoitia. Even inside the same genus, the GC content can vary wildly from 40% in *Plasmodium coatneyi* to 17% in *P. gallinaceum,* as shown in Figure 1.

## 2. Materials and Methods

The dataset of genomes used in this study was obtained from the public databases of GenBank and EuPathDB and consisted of sixty-four Apicomplexa genomes belonging to fifteen different genera, as shown in Table 1 (see also Appendix A for detailed information). The total number of nucleotides from all the genomes analyzed was 1.82 Gb.

We ran RepeatModeler version open-1.0.9 [14] with default parameters for each genome and also for all the genomes merged together as a single FASTA file, to increase detection power of possible low frequency elements that may be present in multiple genomes. A general workflow of the steps used in this work is shown in Figure 2.

Many of the models generated by RepeatModeler may be redundant, either because the software sometimes creates similar models [15], or because the same model may have originated from different genomes. To avoid this redundancy, we clustered the sequences using BLASTn version 2.12 [16], with the parameters -evalue 1 × 10^−10^ and -dust no, to avoid filtering low complexity sequences. We kept the longest model from those which shared more than 90% identity over 90% of the sequence, as the longest models are probably the most complete ones.

Then, we screened the remaining models in each genome with RepeatMasker version 4.1.0 [17] and default parameters, to identify those models that followed patterns associated with conserved genes, such as those with only one or two copies in multiple genomes. Another filtering step was to discard models that emerged from multi-copy protein-coding genes. To do so, we created a database extracting the nucleotide sequences of annotated exons from those apicomplexan genomes with a clear annotation of housekeeping genes or multigene families. We compared our TE models with the exon database using BLASTn with -evalue 1 × 10^−10^ and -dust no. We also ran Tandem Repeat Finder version 4.09 [18] with default parameters against the TE models to identify models with significant fractions of simple repeats. Consequently, models with simple repeats comprising more than 40% of their length were filtered out. For identifying tRNA genes and possible tRNA-derived SINE candidates, we used tRNAScan version 2.0 [19] with default parameters.

We ran the LTRpred version 1.1.0 [20] workflow to identify potential LTR transposons in all the genomic sequences. This software takes into consideration the structure of LTRs and also aims to identify ORFs associated with TE proteins. For identifying non-LTR TEs, we ran MGEScan non-LTR version 3.0 [21] on all the genomes. This software uses probabilistic models based on profiles of hidden Markov models, to identify conserved protein domains associated with retrotranscriptases and apurinic endonucleases, belonging to non-LTR TEs. Both tools were used with the intention of recovering full length LTRs and non-LTR TEs, that may be present in low numbers, but were not identified when the consensus sequences of the models were built with RepeatModeler.

In each of the previously described steps, we discarded models not associated with TEs; this allowed us to reduce the dataset of TE candidate models, making the manual curation easier. We looked for TE evidence by comparing the remaining TE models with the Dfam database version 3.1 [22], using HMMR version 3.1b2 [23] with default parameters. We also extracted the nucleotide sequences of the TEs previously reported by Böhme et al. [6], Templeton et al. [7], and Reid et al. [9], and compared them with our TE models using BLASTn with the parameter -evalue 1 × 10^−20^.

Then, we translated the nucleotides from the good candidate TE models into the six reading frames to identify ORFs associated with TEs. To do this, we used all the translated sequences as queries against the Swiss-Prot database (release 2021_01), using BLASTp with default parameters, except an evalue equal to 1 × 10^−10^. To identify protein domains and signatures from the proteins, we ran these translated sequences against the InterPro collection of databases, in particular Pfam, Gene3D, CDD, PANTHER, SUPERFAMILY, and ProSite, using InterProScan version 5.46.81 [24]. Using these profiles and patterns to identify the presence of TE-related ORFs is extremely useful, as it is a strong signature confirmation of proteins associated with TEs that may not have otherwise been identified. The results obtained from these analyses were manually curated to avoid low quality mapping or spurious hits, such as hits to low complexity or repetitive regions. We also replicated the analyses from Durand et al. [5] by using WU BLAST [25] to scan in our models for TE candidates, using the retrotranscriptase sequence proposed in their work. Finally, we ran TEclass [26] that uses machine learning strategies to help in the classification of our TE models.

After these analyses, we kept only those TE models where there was enough support from Dfam, InterPro, Swiss-Prot, or previously annotated TEs. These TEs were screened in all the Apicomplexa genomes studied using RepeatMasker with parameters -s -e *crossmatch* -lib. The parameter -s indicates the program to perform a more sensitive slow search, -lib was used to indicate that a custom library of candidate TE models was being used, and -e *crossmatch* was used as a search engine. At the end, we used the search results to assess the presence or absence of TEs in each genome, along with the type and number of copies.

The manual curation of the alignment of the TE models involved extracting the hits of the TEs to the genomes and looking for spurious results, such as alignments to parts of the TE models that contain regions with low complexity sequences. We also carefully compared the results from Dfam with those from InterPro; for example, if a result from Dfam showed a hit and then the InterPro results were from a protein domain not associated with TE proteins, such a model was discarded. This strategy aimed to avoid false positives, and could be considered too strict, as it is possible that some TEs in Apicomplexa are extremely divergent and undetectable with our current bioinformatics tools and knowledge about TEs.

## 3. Results

### 3.1. Building and Filtering TE Models

We ran RepeatModeler on each of the sixty-four genomes that total 1.82 Gb and also on a single file with all the genomes merged together. From all these runs, we obtained 10,037 models of consensus sequences using a ‘de novo’ approach, meaning that no library was used as the current knowledge of TEs in Apicomplexa is limited to only a handful of elements. Interestingly, 9508 models came from a single genome analysis and 529 additional models were obtained using a concatenated genome approach. The models obtained initially do not represent confirmed TEs, but due to the nature of the algorithm used by RepeatModeler, just represent the consensus sequences of interspersed repeats, some of which may be potential TEs.

With such a large number of candidate models, we decided to use multiple filtering steps to discard false positives such as genes with multiple copies or simple repeated sequences. We built a Sankey plot using Python version 3.6.9 and Plotly version 4.9.0 (https://plotly.com/, accessed on 5 April 2022) showing how the initial dataset was filtered (see Figure 3). To reduce the number of models to study and to avoid redundant models, the first step was to cluster the sequences of all the candidate models that were at least 90% identical over 90% or more of their sequence, keeping the one with the longest sequence as representative of the cluster. After clustering, 1929 models were discarded, and the dataset was reduced to 8108 sequences. These sequences served as a reference library for a RepeatMasker search against all the individual genomes. We counted the occurrences of each model-related sequence in each genome and manually inspected the quality of the hits to discard spurious results. Some of these models were discarded if they aligned poorly, either if they were found less than three times in each genome or not in their full length, as some models were built with a lax consensus or several similar models were built but one of them aligned better than the others. In the output of RepeatModeler using all the genomes together, we found that many models were just copies of genes conserved across multiple genomes, and consequently, one or two copies were present in a single genome. With this criterion of filtering, we discarded an additional 3355 models and the remaining dataset consisted of 4753 sequences.

Multigene families are abundant in apicomplexan genomes, such as *var*, *rif,* and *stevor* gene families in *P. falciparum*, or *sag*, *esf1*, and *esf2* in species of Eimeria [27]. To discard models that arose from the consensus sequences of conserved genes or those with multiple copies, we created a FASTA file with the exons from all the genomes with available annotations and compared them with the nucleotide sequence from the models. The criteria used were that if the models aligned with more than 40% of their length against the annotated exons, not considering hypothetical proteins or genes that are candidates for being TEs, then we discarded these models. Using this strategy, we identified 771 models that aligned with exons, and consequently, the dataset was reduced to 3982 models.

Although TE ‘de novo’ tools have steps in their pipelines to avoid reporting tandem repeats, on occasions they may fail to detect imperfect tandems, so we ran TRF on all the models and if more than 40% of a model consisted of tandem repeats, it was considered a non-TE repetitive sequence and discarded. The filtering steps for the models were not too stringent as to avoid inadvertently discarding some potential TEs candidates, but at the same time, they were useful in discarding repetitive and low complexity sequences. At this step, we discarded 201 models and 3781 TE candidate models remained for further analysis. Subsequent to all the filtering steps, we obtained a better curated dataset of 3781 models that simplified the manual inspection of the results in the consecutive analysis.

### 3.2. Identification and Classification of TEs

We created a FASTA file from previously reported TEs from Apicomplexa and compared our filtered models with them using BLASTn. This allowed us to identify models similar to previously reported ones (see Appendix A). We were able to identify LTR Gypsy-like elements in *A. taiwanensis*, similar to the results of Templeton et al. [7]. From the genus Plasmodium, we also found LTR Gypsy-like elements in *P. gallinaceum* and *P. relictum*, as reported by Böhme et al. [6]. Intriguingly, none of the other species of Plasmodium have these TEs. Moreover, LTR Gypsy-like TEs are apparently common to all Eimeria species, and we obtained similar results to those reported by Reid et al. [9] in seven species of Eimeria, namely *E. tenella*, *E. necatrix*, *E. mitis*, *E. brunetti*, *E. praecox*, *E. maxima*, and *E. acervulina*, and in our study, all nine species shared the same family of TEs, adding *E. falciformis* and *E. nieschulzi* to the previous list [9]. Additionally, using WU BLAST [25], we scanned our models for a signature of a retrotranscriptase, using the consensus sequence proposed by Durand et al. [5], but we were not able to identify it in our models. However, this is not unexpected, since the strategies used to build TE models were quite different.

Moreover, we compared the candidate sequences with the Dfam database using HMMER [23], and we initially identified 274 models as TE candidates. We manually curated these candidates as in some cases the alignments were too short, or the alignments were in the low complexity regions or satellite sequences. We also used custom plotting Python scripts to visualize the patterns of the alignments, as shown in Figure 4, helping the manual curation of the results. We identified 70 TE models that were good TE candidates, as reported in Appendix A; all of these models had hits to Gypsy TEs. Nevertheless, the results are quite diverse, as most hits are to different Gypsy families, as it is to be expected from very divergent and fragmented TEs (see Figure 4 for an example).

The Dfam database includes models of SINE TEs, such as those derived from 5S RNA, 7SL RNA, and tRNAs, but for an extra confirmation of our results, we also ran tRNAScan against all the models. This tool predicts tRNA genes and pseudogenes, and we were interested in those reported as pseudogenes, as they are potential candidates for tRNA derived SINEs. From the output of tRNAScan, we identified 23 models containing only intact tRNA genes or tRNAs, along with conserved regions across genomes that exceeded by far the length of SINEs.

The nucleotide sequences from the candidate models were translated into their six reading frames to check for any protein similarities, using InterPro databases as a reference. We were especially interested in hits to entries related to TEs, such as retrotranscriptases, ribonuclease H, integrase, aspartyl protease, and gag-pol polyprotein, among others. This allowed us to identify 148 models with protein profiles related to transposons, as reported in Appendix A. Additionally, it allowed us to strictly curate our dataset of TE candidates, as we could identify and discard models with protein domains that are not expected to be related to TEs.

To obtain extra confirmation of the results and to help with the classification of the models, we used BLASTp to compare our models with the curated database Swiss-Prot, allowing us to identify the most closely related TE order and confirm the Dfam results, as some Gypsy/Ty3 TEs are deposited in this database. We also analyzed our models with TEclass as another strategy to help in the classification of TEs and the results are shown in Appendix A. However, we had expected that the classification results from TEclass may not be very precise due to its reliance on previously annotated TEs. Yet, with the extreme GC content of Apicomplexa genomes and very divergent TEs, the software may not be successful in the classification attempts.

For identifying full length LTR TEs, we used LTRpred, which searches the genomes for LTR patterns and looks for ORFs with domains associated to TEs. We obtained 96 TE sequences that aligned with known TE proteins from the Swiss-Prot database, or that contained TE-associated signatures from InterPro databases. The main group of LTR TEs observed were Gypsy-like TEs totaling 83 sequences, but there were also 13 LTR TEs that were characterized due to their internal ORFs associated with TE proteins without a clear identity. This could be due to a great divergence from already known TEs or they may represent new TE families not yet present in the databases.

For searching specifically for non-LTR TEs, we scanned all the genomes using MGEScan non-LTR, which resulted in 46 potential non-LTR TEs. We ran InterProScan using all InterPro databases to confirm these results and we finally identified 44 non-LTR TEs for all the Apicomplexa genomes studied (see Appendix A). A handful of these TEs were found in each genome; seventeen in Eimeria genomes were all classified by MGEScan as LINE R2, three in *B. divergens* were classified as LINE R2, and one of them was a model previously built by RepeatModeler. The classification of seven TEs from *A. taiwanensis*, ten from *G. niphandrodes,* and seven from *C. suis* was not clear, as most of the TEs ended being unclassified or getting mixed results.

We also scanned all the genomes where no TEs were found, searching for TE-related proteins such as retrotranscriptases, transposases, and retrovirus-like proteins, using HMMER with defaults parameters and Pfam HMM profiles to confirm the absence of TEs. The final manually curated dataset consisted of 151 TE models (142 from single genome analysis and 9 from concatenated genomes) that were used as a library in RepeatMasker analysis of all the Apicomplexa genomes. As a result, we were able to calculate the fraction of the genome covered by TEs (see Table 2) and the number of transposons in each genome (see Appendix A).

Intriguingly, we did not find any DNA transposons in the final set of models, and to confirm their absence in the Apicomplexa, we scanned all the genomes for signatures of transposase, a protein that is a hallmark of Class II transposons. We downloaded 123 transposases’ sequences from Repbase and 641 DNA transposon sequences containing transposases and searched in all Apicomplexa genomes using BLASTn with expectation value 1 × 10^−10^, and obtained no results. We also scanned the genomes with 231 transposases from Swiss-Prot using tBLASTn with expectation value 1 × 10^−10^ and identified four transposases. These sequences aligned with full length and high identity to bacterial proteins. As a result, we assumed they were sequence contaminations and discarded them from further analyses. Finally, we searched Apicomplexa genomes for the Class II transposons using the hidden Markov profile approach, one of the most sensitive methods to determine similarities between any sequences. To do so, we employed HMMR and 42 transposases deposited in the Pfam database, and an expectation value of 1 × 10^−10^. Consequently, we only found six transposase candidates, but all of them were also discarded as sequence contaminants. In a similar way, we searched for other DNA transposons, such as Crypton, Helitron, and Polinton, from Repbase using BLASTn. These searches also did not provide any positive results.

### 3.3. Distribution of TEs on the Apicomplexa Phylogeny

Apicomplexa is a group of numerous and extremely diverse parasites with very divergent genomes and their detailed phylogeny has not yet been clearly defined. There are still unresolved issues regarding the branching of some species and even the proper classification into lineages is being disputed. For example, Cryptosporidium was initially placed as a coccidian but later classified as a gregarine [28,29], and *Babesia microti* is quite distant from other Babesia and Theileria, and may even belong to another clade [30]. Here, we propose a tentative phylogenetic tree based on multiple independent studies of different lineages and the relationships among them. We applied the works of Arisue et al. [31], Janouskovec et al. [32], Mathur et al. [10], and Wasmuth et al. [12] to guide the construction of a tentative phylogenetic tree for all the species that we studied. This approach can be considered a reasonable approximation to the actual phylogeny of the species and is useful for the analysis of our data (see Figure 5).

We observed LINE-like transposons in the closely related genera of Babesia and Theileria, with a scattered pattern of distribution among the species studied. In the genus Babesia, we found LINE-like transposons in *B. ovata* and *B. bigemina*, two closely related species that in the assumed phylogeny branched together; they also have remnants of unclassified retrovirus-like proteins. Interestingly, in *Babesia bovis*, that shares the closest common ancestor with both, no TEs were detected, but in *B. divergens*, that branched before *B. bovis*, there are also copies of LINE-like transposons. We also observed the presence of these transposons in *T. equi*, the earliest branching species of Theileria, but not in other members of this genus. No TEs were found in *C. felis*, *T. parva*, *T. annulata*, and *T. orientalis*.

Since *T. equi* shares a common ancestor with the other species of Theileria and Babesia, it can be assumed that this family of LINE-like transposons was present in a common ancestor of both genera, and then lost during the radiation of Theileria. This family was also lost in two other lineages, one in the ancestor of *C. felis*, *T. parva*, *T.annulata*, and *T. orientalis*, and the other one in the lineage of *B. bovis*. An alternative scenario involves an invasion of TEs in two separate events, one in *T. equi* lineage and the other one at the ancestor of *B.ovata*, *B.bigemina*, *B.bovis*, and *B divergens*. This would be followed by TE loss in *B. bovis*. Based on our tree, both scenarios are equally parsimonious; the first involves one event of gaining TEs and two losses, and the second scenario involves two events of gaining TEs and one loss.

In Plasmodium, we found Gypsy-like LTR transposons in *P. gallinaceum* and *P. relictum*, two closely related species, suggesting colonization of their common ancestor. Interestingly, this TE also appears in *H. tartakovskyi*, a distant relative that shares the common ancestor with all the species of the genus Plasmodium, suggesting that gaining these TEs was an independent event. We also found more Gypsy-like LTR transposons in all the nine genomes of Eimeria studied, and in the closely related *C. cayetanensis*, suggesting that an ancestor of this genus was invaded by these TEs, and they were maintained during the radiation of these species. It is interesting that in the closest relatives of Eimeria, the family Sarcocystidae, only the earliest branching *C. suis* has the same type of Gypsy-like TE as Eimeria, but we found no TEs in *H. hammondi*, *T. gondii*, *N. caninum*, *B. besnoiti*, and *S. neurona*. This could be another independent event of *C. suis* gaining TEs, similar to the ancestor of Eimeria. Alternatively, the ancestor of *C. suis* was invaded by these TEs and then they were lost in the lineage of Sarcocystidae. Both scenarios require two events, either two independent gains of TEs in Eimeria and *C.suis*, or a gain in the ancestor of both and a loss in the early radiation of Sarcocystidae.

The gregarines are a more distant group of organisms that are the second earliest to branch off in the whole phylum after Cryptosporidium. We studied two species from this group, *A. taiwanensis* and *G. niphandroides*, and both harbor Gypsy-like LTR transposons, suggesting that a more recent common ancestor of both was colonized by the TEs. Interestingly, no TEs were found in any member of the Cryptosporidium group, even though we studied twelve species from this genus, and not a single candidate model was reported in any of these genomes. To confirm this pattern, we scanned all these genomes with Pfam HMM profiles, but we did not find any potential transposons. As a test, we also tried merging all the genomes together to increase the visibility of TEs that could be present in low numbers, but all the models reported were just conserved genes among species. Consequently, we can conclude with a high confidence that this genus is devoid of TEs.

## 4. Discussion

We observed that the number of TEs in the phylum usually comprises less than 5% of the genome, and there is an apparent absence of TEs in some lineages. In this work, we report for the first time TEs in *B. divergens* and *E. nieschulzi*. The presence of TEs in Apicomplexa was originally reported in *E. tenella* [8], and later confirmed in a comparative genomics study of seven species of the genus Eimeria [9] and in *E. falciformis* [33]. TEs were also identified during the annotation of *A. taiwanenesis* [7], and later in *C. suis* [34] and in *C. cayetanensis* [35]. In a comparative genomics study of Plasmodium species, TEs were found in *P. relictum*, *P. gallinaceum*, and in *H. tartakovskyi* [6]. Despite the reported presence of TEs in *S. neurona* [36], we were not able to confirm TE presence in this species. Initially, some of our models aligned with sequences from Dfam, but these models were discarded after a manual curation because they aligned with low complexity regions. We also had no InterPro or Swiss-Prot results that could confirm the presence of TEs. The previously reported TEs were based exclusively on RepeatModeler results, and it is possible that they were false positives, or we were not able to build models of these TEs due to their small numbers or low sequence conservation. Despite using several data mining methods, we could not find any traces of DNA transposon-related genes in any analyzed genome. Consequently, we can conclude that Apicomplexa were not invaded by DNA transposons.

After building a tentative phylogeny, we found that TEs follow a scattered distribution, particularly in Theileria, Babesia, and Plasmodium, and multiple events of gain and loss are required to explain this pattern. Even if we consider that we missed some TEs, the distribution of Gypsy elements in Plasmodium is restricted to two phylogenetically close species and is absent in the rest of the genus. In Theileria and Babesia, the LINEs found in our models also appear to follow a dispersed pattern.

We can speculate that apicomplexans, as intracellular parasites, experience an evolutionary pressure to maintain a more streamlined and efficient small genome. It has been observed that there is a correlation between genome size and the number of TEs [37]. However, in some cases the lack of TEs may be a reflection of a process of genome reduction that is not uncommon in intracellular parasites, and it has been described in some apicomplexans [38]. Genome reduction can happen in two ways, as a process of gene loss where the parasite becomes more dependent on the metabolism of the host, or as a process that involves contraction of intergenic regions and introns, without gene loss, which is revealed as gene-dense genomes [38]. When the genome of *C. parvum* was first sequenced, it was observed that its gene density almost doubles the one of Plasmodium [39], and it could also be a reflection of the small genome size of around 9 Mb, compared to Plasmodium that is between 20–30 Mb. This may explain the lack of TEs in the genus Cryptosporidium but does not explain larger genomes in other genera.

It was suggested that a decrease in the density of transposable elements could be related to lower GC contents, as it could be possible that there are mechanisms against the invasion of TEs that resulted in the extreme compositional bias that is observed [5]. In our study, if we consider the median GC content of all the genomes studied, that is 33.4%, only 3 out of 32 genomes with a GC content below the median possess TEs. However, 17 out of 32 genomes with a GC content higher than the median were invaded by TEs. Interestingly, the two genomes with TEs and a GC content below the median are *P. gallinaceum*, that has the lowest GC content of only 17%, and *P. relictum*, with the GC content of 18.31%, one of the lowest in the genus Plasmodium. It is also important to consider that possibly the bioinformatics tools that we have at our disposal today are not suited to identify TEs in genomes with such an AT-biased composition.

It has been reported that proteins related to housekeeping genes are significantly divergent from other eukaryotes, making the annotation quite difficult [12]. This situation becomes clear when looking at the annotation in these genomes, where there is a large proportion of proteins annotated as “hypothetical.” For instance, the genes without proper functional assignment represent 92% of *P. fragile* annotation, 81% of *P. inui*, and 65% in *T. parva*. There is a possibility that the apparent absence of TEs in most of these genomes is related to the extreme divergence of the TEs that may hinder identification by sequence similarity, or that these genomes have TE elements that belong to families not yet described.

Interestingly, despite using a number of very sensitive methods, such as HMMER that uses profile hidden Markov models, we could not find any traces of Class II transposons in Apicomplexa. However, we have to take into account that the applied methods were not sensitive enough to detect DNA transposons or their remnants, especially if those transposons evolved unusual structures and their molecular machinery diverged beyond recognition. Although we obtained a number of MITE candidates in the initial scan, all of them were removed as they were most likely to be false positives. This conclusion was supported by the lack of transposase signature in any of apicomplexan genomes.

## 5. Conclusions

We studied TEs in sixty-four Apicomplexa genomes using a strict classification pipeline and looked for evidence that supports our findings in multiple databases. We confirmed TE presence in twenty genomes, and additionally, we discovered TEs in *B. divergens* and *E. nieschulzi*. In general, we observed low numbers of TEs in apicomplexan genomes and a scattered pattern of distribution of TEs on the phylogeny, which can be explained as multiple events of gain and loss of TEs. The genus Eimeria is the only consistent one, where all the members carry LTR Gypsy TEs. The same pattern is also observed in Gregarines, albeit we only had two genomes available to study. In Plasmodium, we only found Gypsy TEs in two closely related species, and in Theileria and Babesia, LINE TEs were found following a scattered pattern. The genus Cryptosporidium is apparently devoid of TEs.

It is possible that the limited presence of TEs is due to evolutionary mechanisms related to their parasitic lifestyle or their unique biology. However, we also have to consider that the low number of TEs we found may be due to the limitations of our bioinformatics tools and of databases to deal with quite complex genomes, with a compositional bias, and extremely divergent genes. We strongly believe that more research is needed and new TEs are yet to be found. Nevertheless, the presented work should be a good starting point in the understanding of the Apicomplexa mobilome, and consequently, better understanding of their genomes’ evolution.

## Figures and Tables

**Figure 1 genes-13-00887-f001:**
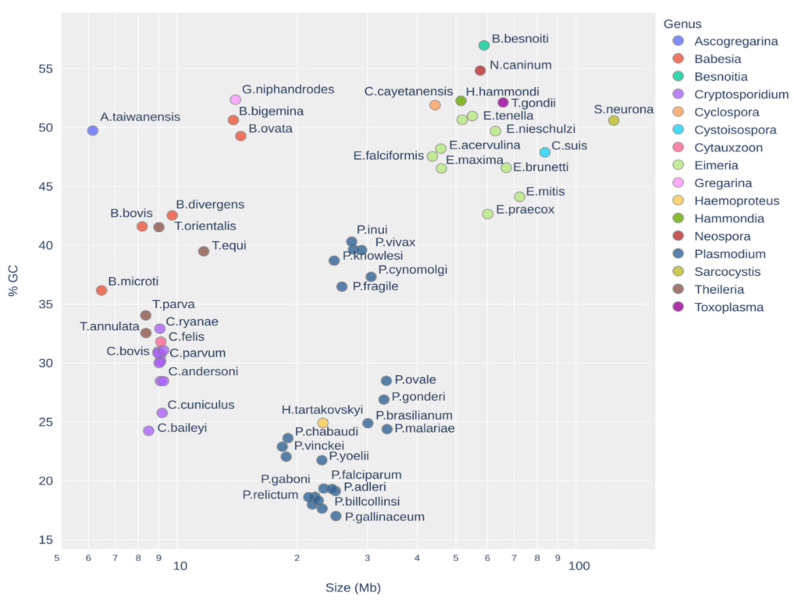
Plot of GC content and genome size (in log scale) used in this work shows the wild diversity of Apicomplexa genomes. For *A. taiwanensis,* it is estimated that the real genome size is approximately four times larger [7].

**Figure 2 genes-13-00887-f002:**
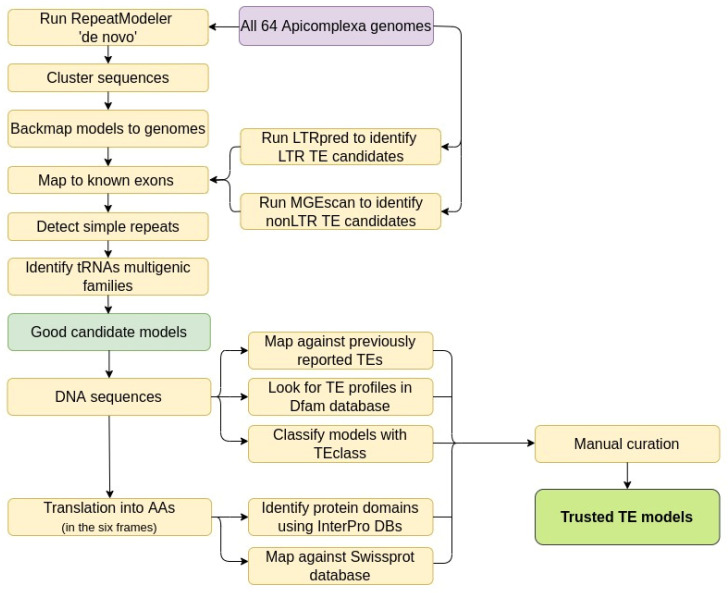
Flowchart illustrating the steps used to filter all the TE candidates and then analyze the sequences, looking for evidence to classify them as trusted TE models.

**Figure 3 genes-13-00887-f003:**
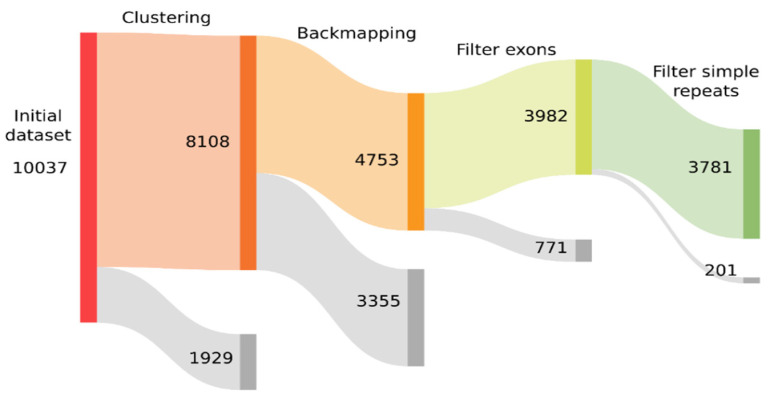
Sankey plot showing the process of filtering of the initial consensus sequences built by RepeatModeler. In each step, in grey is shown how many models were discarded due to the filtering process and in color, how many remained for the next step.

**Figure 4 genes-13-00887-f004:**
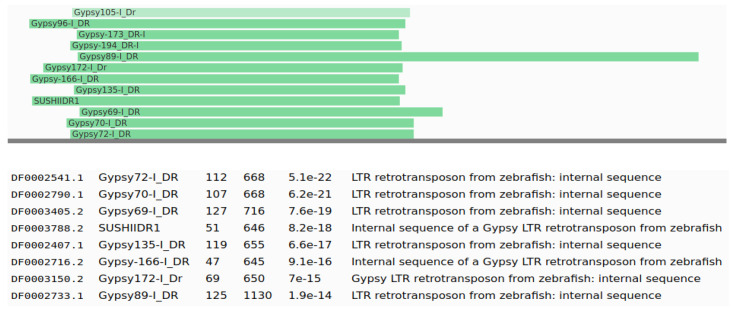
Plot of Dfam alignments from a TE model (internal name Csui_35) from *Cystoisospora suis* that has hits with Gypsy transposons from the Dfam database. The green lines represent the position and length of the hit, with the original sequence as the lower line in grey.

**Figure 5 genes-13-00887-f005:**
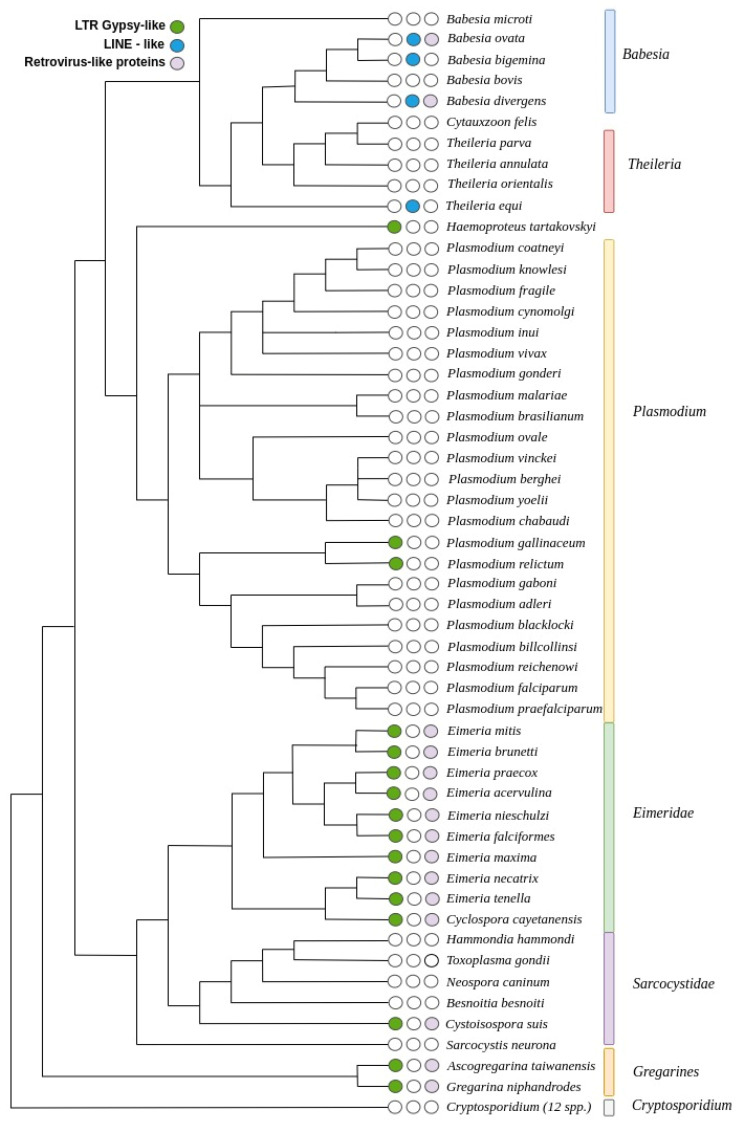
Proposed Apicomplexa phylogenetic tree based on Arisue et al. [31], Janouskovec et al. [32], Mathur et al. [10], and Wasmuth et al. [12], showing the distribution of the main TEs in this phylum.

**Table 1 genes-13-00887-t001:** Number of species used in the presented study, according to their phylogenetic classification based on NCBI Taxonomy [13].

Genus	Number of Species
Ascogregarina	1
Babesia	5
Besnoitia	1
Cryptosporidium	12
Cyclospora	1
Cystoisospora	1
Cytauxzoon	1
Eimeria	9
Gregarina	1
Hammondia	1
Haemoproteus	1
Neospora	1
Plasmodium	23
Sarcocystis	1
Theileria	4
Toxoplasma	1

**Table 2 genes-13-00887-t002:** Percent of the genome covered by our TE models and main family of TE observed after mapping our TE models against all apicomplexan genomes using RepeatMasker. From all sixty-four apicomplexans, we obtained hits to only twenty genomes.

Species	Assembly Genome Length (nt)	TE Coverage (nt)	% of Genome with TEs	Main TE Family
*Ascogregarina taiwanensis*	6,149,411	97,453	1.58	LTR Gypsy-like
*Babesia bigemina*	13,840,936	133,787	0.97	LINE-like
*Babesia divergens*	9,725,408	41,125	0.42	LINE-like
*Babesia ovata*	14,453,397	77,825	0.54	LINE-like
*Cyclospora cayetanensis*	44,363,576	2,176,795	4.91	LTR Gypsy-like
*Cystoisospora suis*	83,637,532	1,169,252	1.40	LTR Gypsy-like
*Eimeria acervulina*	45,830,609	783,904	1.71	LTR Gypsy-like
*Eimeria brunetti*	66,890,165	1,567,917	2.34	LTR Gypsy-like
*Eimeria falciformis*	43,671,268	256,248	0.59	LTR Gypsy-like
*Eimeria maxima*	45,975,062	470,783	1.02	LTR Gypsy-like
*Eimeria mitis*	72,240,319	1,298,206	1.80	LTR Gypsy-like
*Eimeria necatrix*	55,007,932	920,585	1.67	LTR Gypsy-like
*Eimeria nieschulzi*	62,832,469	2,365,007	3.76	LTR Gypsy-like
*Eimeria praecox*	60,083,328	1,423,429	2.37	LTR Gypsy-like
*Eimeria tenella*	51,859,607	637,259	1.23	LTR Gypsy-like
*Gregaina niphandrodes*	14,009,070	432,988	3.09	LTR Gypsy-like
*Haemoproteus tartakovskyi*	23,209,007	12,683	0.05	LTR Gypsy-like
*Plasmodium gallinaceum*	25,034,007	1,351,686	5.40	LTR Gypsy-like
*Plasmodium relictum*	22,607,426	1,105,627	4.89	LTR Gypsy-like
*Theileria equi*	11,674,479	101,054	0.87	LINE-like

## Data Availability

Data are available in a publicly accessible repository: https://github.com/IOB-Muenster/TEs_in_Apicomplexa (accessed on 5 April 2022).

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
