# Peer review of "Mobilome of Apicomplexa Parasites"

_genes, 2022, doi:10.3390/genes13050887_

Round 1

Reviewer 1 Report

Rodriguez and Makalowski performed a comprehensive analysis for the identification of transposable elements (TEs) in the public assemblies of 64 species of Apicomplexa. Almost all species of Apicomplexa are endoparasites of animals, and the genome sizes are highly diverse. Identification of TEs has lagged behind the recent increase in the genome assemblies published, and I agree with the importance of attempting to identify novel TEs from diverse species. They used a new approach for discovering TEs by combining RepeatModeler, LTRpred, and Dfam for TE search and tRNAscan, Blast, etc. for exclusion of false positives. Eventually, they constructed over 150 TE consensus sequences, and the RepeatMasker analysis revealed that Apicomplexa harbors a very small proportion of TEs, at most 5.4% of the genome. 
The results are important, and many genome researchers will be interested in the method and results. However, I have a few concerns listed below, which the authors need to address. 

1. Page 3, lines 82-84; "... and also for all the genomes merged together as a single FASTA file to increase detection power of possible low frequency elements that may be present in multiple genomes."

This seems a good idea. However, it is unclear how this merging approach had an effect on detecting novel TEs. Among the 3781 TE candidates, how many sequences were discovered from the species-specific RepeatModeler analysis? How many were found in the merged genome data only? And, how many were found from both? The authors should provide these numbers. In addition, TEs with a small copy number in Table S4 might be discovered from only the merged genome, which the authors may discuss in the Discussion section. 

2. Page 5, lines 135-136; "After these analyses we kept only those TE models where there was enough support either from Dfam, InterPro, SwissProt or previously annotated TEs."

This means that non-autonomous TEs lacking protein-coding domains, such as SINEs and MITEs, are not covered in this study. If the authors intended to collect models of autonomous (protein-coding) TEs, this should be stated in the text. This is because readers may expect the TE collection in this study to include both autonomous and non-autonomous TEs.

3. Page 4, lines 117-119; "We also extracted the nucleotide sequences of TEs previously reported by Böhme et al. [6], Templeton et al. [7] and Reid et al. [9] and compared with our TE models using BLASTn with the parameter -evalue 1e-20."

I do not know how many and what kinds of TEs have been previously reported in Apicomplexa. This knowledge should be described in the Introduction section, and the authors should discuss whether or not all the previously-reported TEs have been identified in this study. 

Reviewer 2 Report

The MS titled as 'Mobilome of Apicomplexa parasites' by authors of Matias Rodriguez et al., is very interesting, however, the whole MS has to be improved from methods to results, and conclusion.

Major concerns: 

  1. The Title designated as Mobilome of Apicomplexa parasites, however the the TEs presented are restricted to some retrotransposons, LTR, SINE, it is not clear there is no DNA transposons families, or not detected. Aslo  information for LINE detection and annotation is very limited. The MS needs to revise and focus on LTR, or add more analysis of DNA transposons, and LINE.
  2. The methods used are incompltete, and need to improve, the RepeatModel can be used for TE estimation, however it is not fit for de novo TE mining, LTRpred is okay, but I would like suggest to use LTRharvest for LTR, MGEscan-NonLTR for LINE, and do TBLASTN search the DNA TE against by using all DNA TE transposase sequences from RepBase. The EDTA program is a good choice (PMID: 33900591DOI: 10.1007/978-1-0716-1134-0_4) for de novo TE mining.
  3. The abstract needs to improve and add substantial results.

Round 2

Reviewer 2 Report

The MS has been improved significantly, and most concerns have been addressed, and new LINE elements were added, which is substantial different from previous results. However, the annotaion DNA transposons in methods, results and conclusion were not clearly stated. I suggest renamed the title as Retrotransposons in Apicomplexa parasites, and removed for DNA TE section, or try to explain the annotation of DNA TE (class II) clearly in methods, show convicing results in results and conclusion.

Author Response

We believe that we performed adequate DNA transposon analyses and it is an integral part of our project. Therefore we are not willing to remove this part from the manuscript. Moreover, we think that doing so we would give a false impression to the community that DNA transposons might be present in Apicomplexa. However, per this reviewer's suggestions we added a paragraph on cass II transposons in the REsults section (lines 295-311) and another one in the Discussion section (lines 392-395).